# Velocity-based training assessment: Effect of Extended-Relative-Phase-based extracted features on classification

YeongJu Woo
dndudwn31@jnu.ac.kr
Chonnam National University
Buk-gu, Gwangju, Republic of Korea

Hang Thi Phuong Nguyen
hang_208258@jnu.ac.kr
Chonnam National University
Buk-gu, Gwangju, Republic of Korea

HieYong Jeong
h.jeong@jnu.ac.kr
Chonnam National University
Buk-gu, Gwangju, Republic of Korea

## ABSTRACT

Velocity-based training is one of the essential training methods that helps improve athletic performance by providing immediate feedback to athletes. However, there needs to be more ways to evaluate velocity-based training by analyzing the athlete's entire movement. Thus, this study aimed to verify the effectiveness of a newly proposed Extended Relative Phase (ERP) feature on the velocity-based training assessment by using the coordinates of most major joints using Human Pose Estimation (HPE). The difference between experts and novices was compared in the experiment using the proposed feature. The Relative Phase Angle ($RP_{Angle}$) exists to evaluate the combination of each joint's angular displacement and velocity. However, assessing the consistency of repeated movements and comparing angular displacement and velocity with experts takes work. For this reason, the Relative Phase Distance ($RP_{Distance}$) was proposed as a new feature. The dataset trained and predicted the performance verification, including each joint angle, $RP_{Angle}$, and $RP_{Distance}$. The 1D CNN-based deep learning model for training and prediction was used to compare each extracted feature. As a result, the newly proposed indicator had a good effect on the prediction performance of the velocity-based training evaluation.

## CCS CONCEPTS

• **Applied computing** → *Health care information systems.*

## KEYWORDS

relative phase, velocity-based training, human pose, feature extraction, Deep-learning-based personal training system

**ACM Reference Format:**
YeongJu Woo, Hang Thi Phuong Nguyen, and HieYong Jeong. 2018. Velocity-based training assessment: Effect of Extended-Relative-Phase-based extracted features on classification. In *Proceedings of Make sure to enter the correct conference title from your rights confirmation emai (AIDSH 2024)*. ACM, New York, NY, USA, 6 pages. https://doi.org/XXXXXXX.XXXXXXX

## 1 INTRODUCTION

As strength is vital in measuring an individual's performance in sports, optimizing an athlete's strength capacity is often essential and beneficial [18, 31]. Therefore, coaches evaluate muscle strength before and after prescribing a training program to athletes and use it as a critical indicator to judge the performance of a given program. During previous years, the test of one repetition maximum (1-RM) was often regarded as the gold standard for assessing the strength capacity of individuals under practical environmental conditions. The 1-RM test is the maximum weight that can be lifted by an athlete simply once with the correct athletic form. The 1-RM test is the most commonly used test by strength and conditioning coaches to assess strength capacities and evaluate the effectiveness of training programs[6].

Although using a percentage of 1-RM is often referred to as either the traditional or percentage-based approach to calculating training intensity, Poliquin et al.[25] argued that this method based on fixed 1-RM becomes problematic because of the daily variation of 1-RM. Jovanovic et al.[9] and Zourdos et al.[32] have revealed that 1RM varies greatly depending on the condition of the athlete, showing a difference of ±18% and an overall range difference of 36% when 1RM is measured daily with a percentage-based approach. To solve complex problems, clinicians have started measuring body movement velocity during exercise as a marker of intensity rather than the percentage of 1-RM.

Velocity-based training (VBT) is a method of assessing the strength of a given movement by calculating displacement and time through observation of the bar or body speed. The VBT may generally be considered a method of improving the dynamic strength at higher speeds, but, like the 1-RM test, the VBT is simply an objective way to assess the strength in a given movement. Since force and speed have a linear relationship, the strength of a given movement can be objectively quantified even using speed rather than a percentage of the 1-RM test. Through VBT, athletes can obtain information regarding their performance, and coaches can provide specific feedback.

The most commonly employed sensors for measuring VBT are linear position transducers and accelerometers. O'Reilly et al.[22] have investigated whether a single lumbar-worn IMU could identify deviations of seven commonly observed squats, and Lee et al.[13] classified various squat postures through artificial intelligence using IMU sensors. Woo et al.[29] utilized the gait analysis data of the elderly individuals collected through the IMU sensor and found that the symmetries of the left and right feet were different in walking speeds.

However, two things could be improved in the case of VBT measurements. Initially, most VBT-based studies used sensors, but since the number of sensors used in research is limited, the sensors are attached to parts of the body or exercise equipment for measurement. Thus, VBT-based research has the disadvantage that it is not accessible from the environment, and experimenting is possible only when the environment for the measurement is configured.

Only limited studies have attempted to find a universal method for assessing support training; therefore, a comprehensive body movement analysis is warranted. The coupling of each body part during exercise can be expressed through a relative phase (RP)[7, 28]. Although RP is used to obtain information about the relationship (e.g., angle and angular velocity) between the trunk and lower limbs, there are some limitations to the demonstration of continuous exercise.

## 2 RELATED WORKS

### 2.1 Motion-Capture-System-based training assessment

Motion capture is the process of recording the movement of an object or person, which involves measuring the position and orientation of an object or person in physical space. In commercial motion capture, inertial sensors are standard; an example is the Xsens MVN[26]. Xsens utilizes 17 IMU sensors consisting of a combination of accelerometer, gyroscope, and magnetometer to track 6 DOF at the body's joints. Compared to vision-based motion capture, sensor-based motion capture can reduce space limitations but requires a lot of cost and time to install due to the large number of needed inertial sensors. Therefore, existing studies try to use a small number of sensors even if performance degradation occurs.

In addition to using an inertial sensor, there is also an approach using motion capture using an optical sensor or video. The most common device for optical motion capture is OptiTrack[17], which installs multiple infrared cameras that read information from capture sensors attached to the human body. When the sensors provide a 2D position, the motion capture software calculates it as 3D data. The advantage of this approach is that there are no restrictions on the movements performed, it can track many people, and it is also beneficial for fast movements. However, the sensors have disadvantages, such as needing more data due to markers being covered during operation or capturing in a limited space where the camera is installed.

Single-camera-based Human pose estimation (HPE) has made great strides in overcoming the limitations of sensor or motion capture in recent years. For example, Openpose[4] extracted feature points in real-time regardless of the number of people using only videos or photos through deep learning in 2017. Markerless-based MCS(Motion Capture Systems) were less capable than marker-based MCS in studies requiring the tracking of detailed 3D kinematics or fine movements such as finger tracking. However, VideoPose3D[23], announced in 2019, performed effective 3D reconstruction by applying a model based on dilated temporal convolution to 2D critical points of the image. BlazePose[2], announced in 2020, enabled the model to infer the human pose in real-time, even on mobile devices. These models performed lightweight pose estimation using heatmap and regression. Although markerless MCS can offer great potential for extending the scope of movement analysis outside of laboratory settings in a practical way, there is still the problem of the need for more accuracy where detailed 3D kinematics are required for clinical decision-making.

### 2.2 Human-Pose-Estimation-based training assessment

Human Pose Estimation (HPE) is computer vision technology that predicts a person's posture by specifying a person's joints or essential body parts as key points and is widely used in various fields such as autonomous driving, sports, medical, and metaverse[5, 11, 14, 16, 21]. In addition, as HPE technology advances, the fitness technology market is filled with AI-powered personal trainer apps.

ALFA-AI[1] monitors the user's exercise execution with real-time AI analysis and provides real-time visual and auditory improvement feedback. The user's key joints are tracked through a two-dimensional coordinate system, and personal AI training is continuously adjusted according to the actual user's performance. infiGro[8], created by Infivolve, is a fully automated, AI-powered, digital personal trainer app that guides, analyzes, corrects, and motivates in real time via your phone's camera. It shows an example video of an expert and counts the number of repetitions of the user through pose estimation.

However, as we surveyed, these applications are programmed through simple algorithms such as estimating the angle through the body's coordinate system or whether or not a given threshold value is exceeded. Alternatively, there is a disadvantage that the program proceeds without considering the count of exercises and the condition or situation of the player performing the exercise. Alternatively, since only a two-dimensional coordinate system is used, it is challenging to apply in a wild environment.

## 3 METHOD

### 3.1 Relative Phase

Continuous Relative Phase (CRP) indicates the positional change in coordination by describing phase relationships between the two joints[3, 12]. RP is often measured to quantify the relationship between the kinematics of two mechanically connected joints during a certain period to analyze specific movements, such as human gait. Figure 1 depicts the overall method for calculating the relative phase[20]. To calculate RP, we subtract the phase angle of the proximal segment ($PA$) from the distal segment for each $i^{th}$ data point calculated from the time-normalized phase portrait.

However, since the Relative Phase analyzes the data using only the Phase Angle, it is suitable for temporally analyzing the coordination of two joints. Nevertheless, the Relative Phase's disadvantage is that it is inadequate to explain the strength or stability of motion.

When multiple squat cycles are plotted on the same phase portrait, the amount of variability in the path of the trajectory can be used to qualitatively evaluate the stability of the neuromuscular system under the given exercise-intensity condition. Slight variations in the trajectory are because of the response of the neuromuscular system to global and local perturbations experienced during the squat cycle. Such flexibility enables the neuromuscular system to maintain a stable and proficient movement pattern. Therefore, excessive variability is associated with instabilities in the behavior of

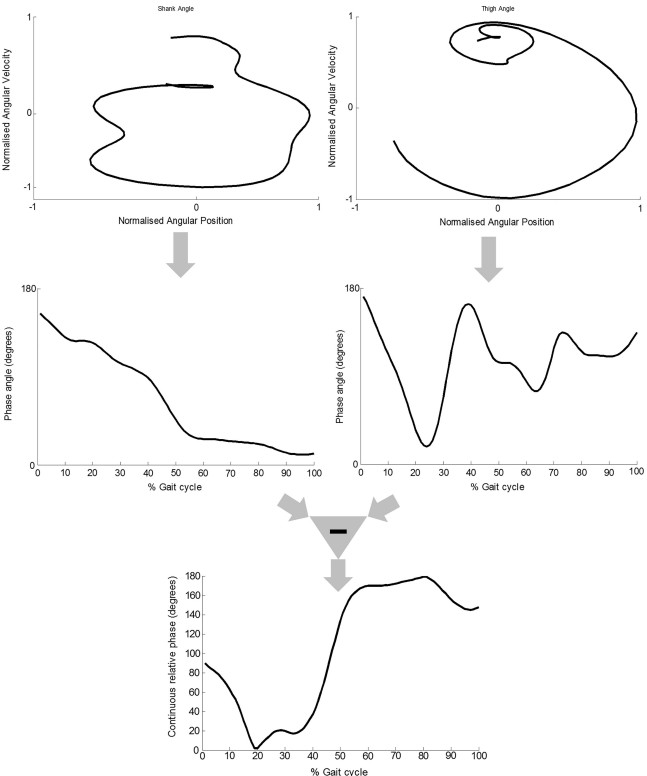

**Figure 1: Calculation of relative phase**

the neuromuscular system. This extreme variability may suggest a lack of control of the multiple degrees of freedom and may indicate disorder in the organization of the neuromuscular system.

Therefore, an Extended Relative Phase is proposed that extends from the existing Relative Phase by adding the Phase Angle and the Phase Distance, which can describe the degree of variability and stability of motion.

## 3.2 Extended Relative Phase

For calculating ERP, we first utilize Blaze Pose, which supports the ML (machine learning) kit pose detection API to detect the person's skeleton in the image. Blaze Pose is one kind of human pose estimation algorithm that infers 33 3D landmarks and background segmentation masks for whole bodies in RGB video frames. Table 1 shows the results of the pose estimation quality of the BlazePose GHUM model[30] used in MediaPipe Pose[19]. The result was evaluated through three different validation datasets: Yoga, Dance, and HIIT. The results confirmed that the used model in this study showed high performance with high accuracy.

The Hampel identifier[24] was used to eliminate the outliers of the estimated 33 3D landmark coordinates. Subsequently, the Savitsky-Golay filter[27] was applied to smooth the coordinates. Only two coordinates of interest, such as the hip and knee, were used to evaluate the squats among the 33 adjusted 3D landmark coordinates and normalize the calculated Angle and angular velocity. For angle normalization, we applied the robust filter, which has

**Table 1: The results of pose-estimation quality of BlazePose GHUM model in MediaPipe Pose**

| Method | Yoga | Dance | HIIT |
|---|---|---|---|
| BlazePose-GHUM (Heavy) | 96.4 | 97.2 | 97.5 |
| BlazePose-GHUM (Full) | 95.5 | 96.3 | 95.7 |
| BlazePose-GHUM (Lite) | 90.2 | 92.5 | 93.5 |
| AlphaPose-Resnet50 | 96.0 | 95.5 | 96.0 |
| Apple Vision | 82.7 | 91.4 | 88.6 |

strong characteristics in outliers, and the MinMax filter, which normalizes the data between -1.0 and 1.0. Through the preprocessing of these data, the value of the normalized Angle was obtained as follows[10]:

$$\theta_i^{rb} = \frac{\theta_i^{raw} - \mathbf{Q}_1(\theta^{raw})}{\mathbf{Q}_3(\theta^{raw}) - \mathbf{Q}_1(\theta^{raw})}, \tag{1}$$

$$\theta_{i'} = \frac{2 \times (\theta_i^{rb} - \theta_{min}^{rb})}{\theta_{max}^{rb} - \theta_{min}^{rb}} - 1 \tag{2}$$

where $\theta_{i'}$ represents the normalized Angle, $\mathbf{Q}$ represents the Quantile range, $\theta_i^{raw}$ represents the original Angle, $\theta_i^{rb}$ represents the Angle to which the robust filter is applied, and $i$ indicates the point of the cycle. Compared with the Angle's normalization, the angular velocity normalized through the MaxAbs filter is as follows:

$$\omega_{i'} = \frac{\omega_i}{|\omega_i|_{max}} \tag{3}$$

where $\omega_{i'}$ represents the normalized angular velocity, $\omega_i$ represents the initial angular velocity ($\omega = \frac{d\theta}{dt}$), and $i$ indicates the point of the cycle.

The Extended Relative Phase with the addition of phase distance, i.e., the radius of the phase, is composed of $RP_{Angle}$ and $RP_{Distance}$ and is calculated by the following expression:

$$PA_i = \tan^{-1}(\frac{\omega_{i'}}{\theta_{i'}}),$$
$$RP_{angle}^i = PA_{hip}^i - PA_{knee}^i \tag{4}$$

$$PD_i = \sqrt{(\theta_{i'})^2 + (\omega_{i'})^2},$$
$$RP_{distance}^i = PD_{hip}^i - PD_{knee}^i \tag{5}$$

where $PA$ describes the Phase Angle, $PD$ describes the Phase Distance, $\omega$ represents the normalized angular velocity, and $\theta$ represents the normalized angular position.

Figure 2 describes used angles and angular velocities and the method of calculating the proposed ERP(extended relative phase) indicator for the assessment of velocity-based training: (a) representation of two joints, (b) description of the results of normalized angles and angular velocities, (c) description of the results of the Phase Portrait of the hip-knee during exercise, (d) representation

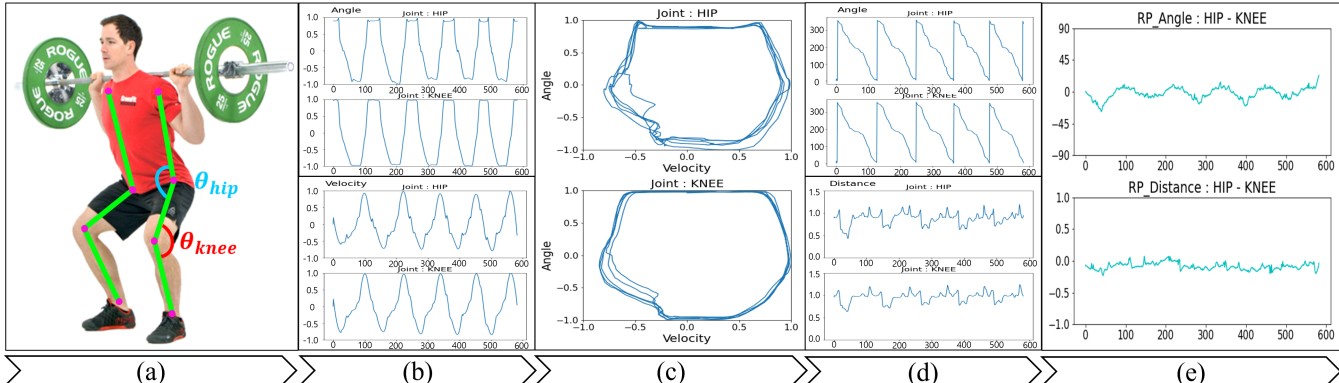

**Figure 2: An explanation of used angles and angular velocities and the process of calculating the proposed ERP (extended relative phase) algorithm for the velocity-based training assessment**

of the results of $PA$ and $PD$, and (e) description of the results of the relative phase of $RP^i_{Angle}$ and $RP^i_{Distance}$, respectively.

## 4 EXPERIMENTAL ENVIRONMENTS AND SYSTEMS

After capturing the user image, the 3D coordinates for each joint are estimated. Subsequently, the proposed ERP algorithm is applied for feature extraction in velocity-based training. Finally, the extracted features classify the difference between experts and users. The back-squat footage of seven non-professional participants was captured to collect the data related to velocity-based training for novices. The footage was captured using a resolution of 1920 × 1080 and a camera of 30 fps in the configured environment. The distance between the camera and the participants was set to 380(± 5) cm for the camera's focus. In addition, the camera's height was fixed to 130 cm, although the participants' heights were slightly different (mean ± standard deviation = 167.1 ± 8.7 cm)

**Table 2: Personal and physical information for each participant**

| Subject | Gender | Age [years old] | BMI [$kg/m^2$] | Exercise Experience |
|---------|--------|-----------------|-----------------|---------------------|
| Sub #1 | M | 27 | 28.37 | 3 years (sports) |
| Sub #2 | M | 27 | 22.60 | 4 years (free weight) |
| Sub #3 | M | 28 | 25.47 | 1 month (free weight) |
| Sub #4 | M | 30 | 22.79 | NA |
| Sub #5 | F | 21 | 23.63 | 7 month (pilates) |
| Sub #6 | F | 25 | 24.56 | 6 month (pilates) |
| Sub #7 | F | 25 | 19.81 | 6 month (pilates) |

Table 2 summarizes the personal and physical information of the seven participants. All participants gave informed consent to include them before participating in the study. We also collected information regarding the athletic careers of the participants to provide reliability for the experiment. The experimental procedures were performed under the Declaration of Helsinki and approved by the Clinical Trial Center Ethics Committee, Department of Medical Innovation, Osaka University Hospital (no. 15408, 11 March 2016).

The participants performed 20 squats in 3 sets with a 15 kg weight bar. The break time between each set was set to 60 seconds, and they tried to proceed with the squats at a constant speed and movement.

## 5 EXTENDED-RELATIVE-PHASED-BASED EXTRACTED FEATURES ON CLASSIFICATION

To verify the effectiveness of these indicators in actual case analysis, we utilized a 1D-CNN-based deep learning classification model. The labels were divided into quartiles according to the exercise abilities of the participants. We divided the models into several categories based on input differences, conducted individual training for each, and analyzed the results. Using 1D CNN for sequence classification has the advantage of directly learning from raw time series data without requiring domain-specific knowledge to engineer input features manually.

### 5.1 Data Augmentation & Pre-processing

We attempted data augmentation through sampling to address the limited amount of data. Considering that the information, such as trends and shapes in the time series data we want to use, is crucial, we judged that common techniques in data augmentation may not be suitable. Therefore, we utilized random sampling to augment angular position $\theta$ and angular velocity $\omega$ for each participant by 1,000 instances. Using the augmented $\theta$ and $\omega$, we calculated $RP_{Angle}$ and $RP_{Distance}$. As the calculated ERP had significantly larger values than $\theta$ and $\omega$, we scaled the ERP between -1 and 1 using the MinMax filter to address data bias. The finalized dataset was split into train and test data in a 7-3 ratio, maintaining the class distribution ratio during the split.

### 5.2 Model Description

The model used for training has the structure shown in Figure 3. In this study, we configured the network framework by adding Global

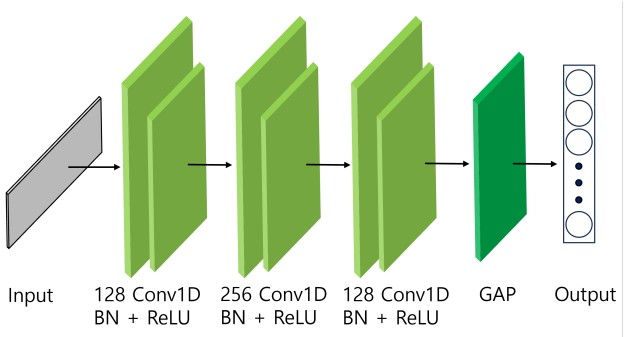

**Figure 3: Model architecture**

Average Pooling (GAP)[15] to the 1D-CNN. To avoid overfitting, dropout is added after each layer. Finally, GAP is applied to reduce the number of output parameters before generating the output. Instead of adding fc-layers on top of the feature maps, we take the average of each feature map, and the resulting vector is fed directly into the softmax layer.

### 5.3 Analysis results

We conducted four experiments by varying the input parameters of the model. In Experiment 1, we utilized only eight features, excluding ERP, using $\theta$ and $\omega$ of both hips and knees. Experiment 2 involved adding the existing $RP_{Angle}$ to the inputs, resulting in 10 features. For Experiment 3, we introduced a new feature, $RP_{Distance}$, to the existing eight features, resulting in 10 features used in the experiment. In the final experiment, we included $RP_{Angle}$ and $RP_{Distance}$, totaling 12 features in the input. We compared the results of these four experiments.

**Table 3: Result of Training**

| Experiment | Features | Val auc & loss | test acc |
|---|---|---|---|
| Ex.#1 (Baseline) | $\theta, \omega$ | [0.7500, 0.7500] [0.7620, 0.8072] | 0.75 |
| Ex.#2 (Comparable 1) | $\theta, \omega, RP_{Angle}$ | [0.7500, 0.7500] [0.4090, 0.4176] | 0.9983 |
| Ex.#3 (Comparable 2) | $\theta, \omega, RP_{Distance}$ | [0.7500, 0.7500] [0.5251, 0.5362] | 0.9913 |
| Ex.#4 (Best) | $\theta, \omega$ $RP_{Angle}, RP_{Distance}$ | [0.7500, 0.7500] [0.0016, 0.0004] | 1.0 |

Table 3 summarizes the features, accuracy, and loss values for each experiment. Furthermore, Figure 4 illustrates the confusion matrix for each experiment in the test dataset. In conclusion, the model incorporating the existing $RP_{Angle}$ (Experiment 2) performed better than Experiment 1, which used only $\theta$ and $\omega$. Furthermore, it was observed that the model trained with the addition of the proposed ERP to the existing features outperformed the models trained with only $RP_{Angle}$ or $RP_{Distance}$ added to the current features.

## 6 CONCLUSION

This study proposes an Extended Relative Phase (ERP) indicator to assess velocity-based training. It utilizes this indicator extracted

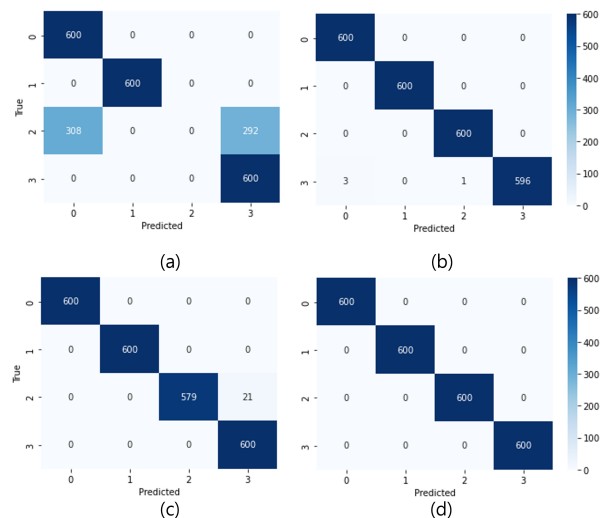

**Figure 4: Confusion Matrix**

from a specific exercise video to demonstrate the difference between experts and novices and verify the validity of the proposed indicator. According to the experiment, we can better understand the relationship between users' exercise performance and ERP. The plot of the conventional $RP_{Angle}$ index provided coordination between the angular displacement and the angular velocity. The plot of the newly defined $RP_{Distance}$ index enabled us to confirm the procedure through which stable periodic motion was performed during intense motion performed by the participants. Furthermore, We validated the newly proposed indicator's performance and validity by comparing it with a model using only conventional indicators.

Our markerless-based research is a groundbreaking method with the advantage of being fast and free from spatial restrictions compared to conventional sensor or motion capture techniques. In specific exercises, coaches can simultaneously evaluate the relationship between the joint temporal coordination and the stability of the exercise intensity through ERP without specific technical knowledge. However, the method has certain limitations, such as being less accurate than sensor-based evaluation and having less internal data in the experiment.

Given the growing interest in fitness worldwide, the indicators and evaluation methods presented in the paper are thought to be a new and objective way to evaluate a player's performance in the fitness field. We expect this method to expand to fitness and various fields, such as patient walking data for rehabilitation and posture data in multiple sports.

## ACKNOWLEDGMENTS

This research was supported by the Basic Science Research Program through the National Research Foundation (NRF) of Korea grant, funded by the Ministry of Education (NRF-2021R1I1A3055210), and partially by Institute of Information & communications Technology Planning & Evaluation (IITP) under the Artificial Intelligence Convergence Innovation Human Resources Development (IITP-2023-RS-2023-00256629) grant funded by the Korea government(MSIT).

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
