# OpenReview forum: "Velocity-based training assessment: Effect of Extended-Relative-Phase-based extracted features on classification"
_KDD.org/2024/Workshop/AIDSH — KDD-AIDSH 2024 Poster_

### Official Review · Reviewer_3ijp · 2024-06-18
**Doubtable experiment results due to very limited dataset size, potentially problematic data augmentation and uncomparable results.**

**Rating:** 3
**Confidence:** 4

**Review:**

This work proposes a new feature extraction technique for markerless velocity-based training assessment. This article suggests a new feature named extended relative phase (ERP) to enhance the evaluation for training in kinematics. I conclude the pros and cons as below:

Pros:

- The proposed feature is derived from the traditional phase indicators to capture a higher level of information with demonstrated examples to show its advantage compared to the traditional indicators.

- The experiment results validate the effectiveness of the proposed ERP feature by an ablation study.

Cons:

- The size of the used dataset is very small with only 7 subjects. Although certain data augmentation technique (random sampling) is used, I doubt if random sampling will help in data augmentation. If the trends and patterns in your time series data are crucial, simple random sampling might not be sufficient. It may not capture the underlying structure and could introduce noise instead of meaningful variation. Random sampling to augment angular position and angular velocity can be useful if these variables follow a predictable distribution, but there is no evidence of this assumption holding. I cannot find the clues to convince me random sampling is a good method in this case.

- I am also confused by the classification results shown in the experiments. What is your predicted target? In the confusion matrix, there are only encoded labels (0, 1, 2, 3), so what do these numbers mean? Really need more discussion and explanation here. And the reported accuracy is so high that can be suspicious (100% of accuracy!!). Just wonder if the authors tried the proposed method on other public datasets to compare with other work.

- Some used symbols are not explained. For example, what is "Q" in Eq. 1? I can guess it should be a function, but what this function does is not explained.

---

### Official Review · Reviewer_j8my · 2024-06-21
**Good results achieved with a model based on relative phase (RP). Even though the novelty of this work seems not sufficient enough and results only have reported on private datasets.**

**Rating:** 5
**Confidence:** 4

**Review:**

This paper proposed a method based on relative phase distance to improve the prediction performance of the velocity-based
training evaluation.

# Quality:
On overall, the paper reads good, but there are many typos which the author should check very carefully.

# Clarity:
The entire presentation of the paper (considering only the main body without any supplementary file) falls between a border line paper and a weakly rejection.

# Originality and Significance:
It is not completely clear that it will be significant. Will it work on complex vision datasets having multiple objects ? That is the question.

# Limitations:
1.The generalization ability of the proposed method cannot be demonstrated. This paper only gives quantitative comparisons on private datasets. The authors should provide more quantitative and qualitative comparisons on real-world public datasets.
2.The main weakness of this paper is that the proposed approach is not technically sound.
Some key details are not explained clearly, and thus it is difficult to understand how it works.
3.Some commonly-used metrics for evaluating model performance, such as Precision, Recall

---

### Decision · Program_Chairs · 2024-06-28

Accept (Poster)